# Superspreading of SARS-CoV-2 in the USA

**Calvin Pozderac**  *, **Brian Skinner**

Department of Physics, Ohio State University, Columbus, Ohio, United States of America

* pozderac.16@osu.edu

## Abstract

A number of epidemics, including the SARS-CoV-1 epidemic of 2002-2004, have been known to exhibit superspreading, in which a small fraction of infected individuals is responsible for the majority of new infections. The existence of superspreading implies a fat-tailed distribution of infectiousness (new secondary infections caused per day) among different individuals. Here, we present a simple method to estimate the variation in infectiousness by examining the variation in early-time growth rates of new cases among different subpopulations. We use this method to estimate the mean and variance in the infectiousness, $\beta$, for SARS-CoV-2 transmission during the early stages of the pandemic within the United States. We find that $\sigma_\beta/\mu_\beta \gtrsim 3.2$, where $\mu_\beta$ is the mean infectiousness and $\sigma_\beta$ its standard deviation, which implies pervasive superspreading. This result allows us to estimate that in the early stages of the pandemic in the USA, over 81% of new cases were a result of the top 10% of most infectious individuals.

**Data Availability Statement:** We use publicly available data taken from the data set provided by the Center for Systems Science and Engineering (CSSE) at Johns Hopkins University. The paper describing the data can be found at: https://doi.org/10.1016/S1473-3099(20)30120-1. The data can be

## Introduction

The temporal growth of an epidemic is often characterized by either a time scale (such as the doubling time) [1, 2] or by the reproduction rate $R_0$, which indicates the average number of new infections produced by each infected individual [3]. Estimates of $R_0$ for the current pandemic of SARS-CoV-2 range from 1.4 to 3.8 [4–7]. Neither of these numbers, however, gives any information about the distribution of infectiousness among individuals—i.e., whether new infections arise relatively uniformly from all infected individuals, or whether new infections are driven primarily by a small number of highly infectious individuals. The latter case is commonly referred to as "superspreading", and different epidemics exhibit superspreading to different degrees. For example, during the outbreak of SARS CoV-1 in 2002-2004, over 80% of cases were observed to result from the top 20% most infectious individuals [8, 9]. Understanding the degree of superspreading in the current pandemic of SARS-CoV-2 is crucial for developing strategies to mitigate continued spread and informing an educated reopening procedure [10–13].

Here we present a simple and direct method to understand how the infectiousness (also called the "reproduction rate" of the disease) varies among infected individuals. At late times after the onset of an epidemic, the number of infected individuals is large, and consequently any statistical fluctuations in the growth rate are relatively small, so that the growth rate is well characterized by the mean infectiousness, $\mu_\beta$. However, at early times, when there are relatively few cases, the growth rate is stochastic and the degree of randomness depends on the variance

found at https://github.com/CSSEGISandData/
COVID-19/tree/master/csse_covid_19_data/csse_
covid_19_time_series under the filename
time_series_covid19_confirmed_US.csv.

**Funding:** The author(s) received no specific
funding for this work.

**Competing interests:** The authors have declared
that no competing interests exist.

in infectiousness, $\sigma_\beta^2$, between individuals (Fig 1a). By examining the variance in growth rate across subpopulations at these early times (Fig 1b), we are able to infer the variation in the distribution of infectiousness. In our analysis we divide the US cases into counties and observe how the variance in growth rate across them evolves as the number of cases increases.

Formalizing this idea, we present a derivation of the variance in the exponential growth rate, or number of new cases per infected individual per day, $\Delta I/I$, using an SIR framework that incorporates a probability distribution for the infectiousness of a given individual. Our result implies a simple method for estimating the mean, $\mu_\beta$, and variance, $\sigma_\beta^2$, of the infectiousness $\beta$. We apply this method to data for COVID-19 cases in the USA, and find a mean infection rate of $\mu_\beta = 0.18$ cases/day and standard deviation of $\sigma_\beta \gtrsim 0.59$ cases/day. Since the standard deviation is considerably larger than the mean, with $\sigma_\beta/\mu_\beta \gtrsim 3.2$, we conclude that superspreading is prevalent. By our estimate, these results imply that at least 81% of new cases are caused by the top 10% of most infectious individuals. Our method, which uses only a direct measurement of variance in detected case data in the USA, is consistent with estimates of superspreading using surveillance data [14], secondary-case data [15], and more complicated estimates of cluster size distribution using Markov Chain Monte Carlo [16].

## Results

### Variance in growth rate in the SIR model

We derive a relation between the variance in the case growth rate and the variance in individual infectiousness between individuals in the population. We start with a standard discrete-time SIR model [17], which is governed by the following difference equations:

$$\begin{aligned}
\Delta S &= -\beta I \frac{S}{N} \\
\Delta I &= \beta I \frac{S}{N} - rI \\
\Delta R &= rI
\end{aligned} \tag{1}$$

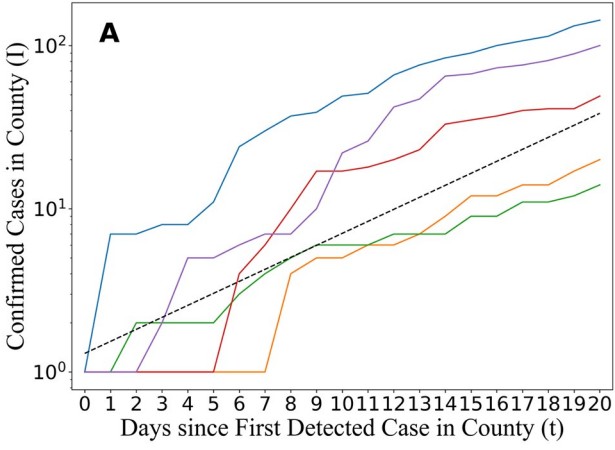

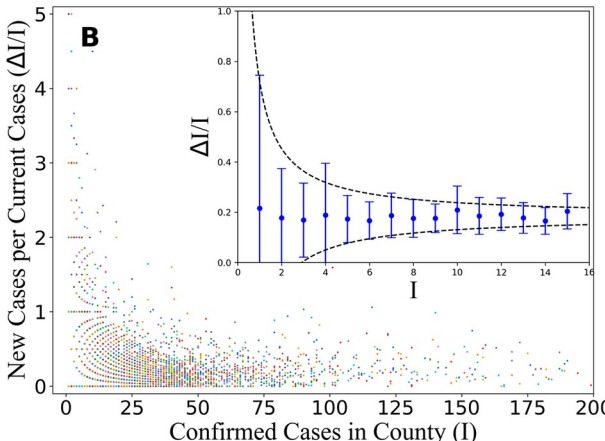

**Fig 1.** (a) Illustration of the variance in early-time growth rate of new cases. At early times, there is noticeable variance in the growth rate between counties. As the number of cases grows, all counties stabilize towards the average growth rate $I \sim (1+\mu_\beta)^t$, (dashed black line) where $t$ is the number of days since the first case in a county. The counties shown are Boulder, CO (blue), St. Mary, LA (purple), Vanderburgh, IN (red), Mesa, CO (orange), and Jones, GA (green). (b) The number of daily infections per infected individual as a function of total infections. In the main figure, each point corresponds to a given county (across all US counties that never report $\Delta I < 0$) at a given time point (within the first 14 days after the first infection reported in that county). As the number of cases increase, all counties converge to the mean infection rate. The mean (points) and variance (bars) of $\Delta I/I$ at a given $I$ are shown in the inset. The variance decreases like $(\mu_\beta + \sigma_\beta^2)/I$ (black lines).

Here, $N$ is the total population and $S$, $I$, and $R$ are the time-dependent numbers of susceptible, infected, and recovered individuals, respectively. The parameters $\beta$ and $r$ encode the infectiousness and recovery rate of a disease within a population. The time is effectively discretized into days by the available data, so we use $\Delta I$ rather than the usual time derivative, $dI/dt$. The SIR description typically assumes fixed values for $\beta$ and $r$ across the population. However, in superspreading contexts there is a substantial variance in the infectiousness within a population [8, 9, 18, 19]. We account for this variation by introducing a probability distribution of infectiousness, $p(\beta)$, so that the probability for a randomly-selected individual to have infectiousness in the range $(\beta, \beta + d\beta)$ is given by is given by $p(\beta)d\beta$.

For an individual with a given infectiousness, $\beta$, the probability of infecting exactly $n$ others in a day follows the Poisson distribution, $\mathrm{Pois}(n;\beta)$. The probability that a randomly selected individual will infect $n$ others is given by combining the Poisson distribution with the distribution $p(\beta)$, giving

$$P(n) = \int_0^\infty d\beta \frac{e^{-\beta}\beta^n}{n!} p(\beta). \tag{2}$$

The first two moments of $P(n)$, $\mu_n$ and $\sigma_n^2$, can be calculated independent of the form of $p(\beta)$:

$$\mu_n = \sum_{n=0}^\infty nP(n) \quad = \mu_\beta \tag{3}$$

$$\sigma_n^2 = \sum_{n=0}^\infty (n - \mu_n)^2 P(n) \quad = \mu_\beta + \sigma_\beta^2 \tag{4}$$

Eq (4) represents the variance, among all infected individuals, of the number of new infections caused by a single person in a given day. When there are $I$ active cases, the mean number of new cases per infected person, $\Delta(I + R)/I$, is given by the average of $I$ random variables drawn from the distribution $P(n)$. By the central limit theorem, it follows that $\mathrm{Var}(\Delta(I + R)/I) = \sigma_n^2/I$. Additionally, in the SIR model with a finite total population $N$, $\Delta(I+R)/I = \beta S/N = \beta(1 - (I + R)/N)$ decreases as the susceptible population continually shrinks. Effectively, $p(\beta)$ is scaled by the factor $(1 - (I + R)/N)$, which represents the fraction of the population that remains susceptible. Consequently, $\mu_\beta \to \mu_\beta(1 - (I + R)/N)$ and $\sigma_\beta^2 \to \sigma_\beta^2(1 - (I + R)/N)^2$. Therefore the total variance in $\Delta(I+ R)/I$ follows:

$$\mathrm{Var}\left(\frac{\Delta(I + R)}{I}\right) = \frac{\mu_\beta\left(1 - \frac{I+R}{N}\right) + \sigma_\beta^2\left(1 - \frac{I+R}{N}\right)^2}{I} \tag{5}$$

This result becomes simpler in the limiting case where there is no significant change in the susceptible population ($N \to \infty$) and no recovery ($R \to 0$). In this limit, we retrieve the case of simple exponential growth, for which [20]

$$\mathrm{Var}\left(\frac{\Delta I}{I}\right) = \frac{\mu_\beta + \sigma_\beta^2}{I}. \tag{6}$$

In the limit $\sigma_\beta \to 0$, where every infected individual has the same infectiousness $\mu_\beta$, the variance in the average infection rate is simply $\mu_\beta/I$, which corresponds to the variance in a Poisson process with rate $\mu_\beta$.

In the case of SARS-CoV-2, it is well established that there are asymptomatic carriers [21–23] who transmit the virus without being detected, as well as other infections that are

undetected or unreported. Current estimates typically predict that only $10 - 25\%$ [24–26] of cases are detected. One can attempt to address this effect by assuming that there is a fixed detection probability, $p_{\text{det}}$, and that the entire infected population, regardless of symptoms, follows the same infectiousness distribution $p(\beta)$. In this case, there are many more infected individuals, $I \sim I_{\text{det}}/p_{\text{det}}$, than those detected, which reduces the statistical fluctuations in the growth rate and makes our calculation of $\sigma_\beta^2$ a lower bound. The effect of undetected cases is considered in more detail in the S3 Appendix. In order to be conservative (especially given the possibility that asymptomatic cases have a lower rate of infection than symptomatic ones [27, 28]), the results we present here use $p_{\text{det}} = 1$.

## Data for COVID-19 in the USA

We now turn our attention to data for total detected cases of COVID-19 in the USA, taken from the publicly available data set at Ref. [29]. In the following analysis we limit our consideration to only a short timescale ($\sim 14$ days) after the first infection is detected in a given county. This limitation in time scale serves three main purposes; first, it is likely that through changes in policy, lockdown, social distancing, mask usage, etc., the average infectiousness within the population is time-dependent. By restricting ourselves to a relatively small window of early times, we may assume that there is a constant average infectiousness. Second, considering only beginning stages allows us to neglect the possible saturation of the susceptible population, effectively allowing us to take the $N \to \infty$ limit. Finally, the recovery period for COVID-19 ranges from 7-14 days [30, 31] and so by considering this two week period, we can treat our system as if there is limited recovery and $R \to 0$. These restrictions allow us to treat the USA data using the exponential case, Eq (6).

In our analysis, the population is divided into geographic regions and the variance is calculated across different trajectories $I(t)$. The US cases are divided by county. For each county, we calculate the average number of new cases per current case per day, $\Delta I/I$, for the first 14 days after the first infection is detected in that county. The variance in $\Delta I/I$ is then calculated among all counties that have a given fixed value of $I$ (we present data only for values of $I$ that have at least 250 corresponding counties). As shown in Fig 2, the US data generally follows the predicted $\sim 1/I$ trend. An unbiased fit of the data gives $\text{Var}(\Delta I/I) \propto I^{-0.74}$. From Eq (6), we calculate $\mu_\beta + \sigma_\beta^2$ by averaging $\text{Var}(\Delta I/I) \times I$, weighted by the number of instances at each $I$ value. One might worry that the main source of variation comes from differing average growth rates, $\mu_\beta$, in various counties (e.g. rural vs. urban). However, we show in the S2 Appendix that variance in $\mu_\beta$ across counties is too small to explain the large observed variance in $\Delta I/I$.

We calculate $\mu_\beta$ from the entire USA population by averaging all values of $\Delta I/I$ weighted by the current number of infections. Equivalently, we sum the number of cases caused each day and then divide by the sum of the number of cases across those days. This procedure gives the mean infectiousness, $\mu_\beta$, and thus from Eq (6) and the fitted slope in Fig 2, we can infer $\sigma_\beta^2$.

This calculation yields $\mu_\beta = 0.18$ cases/day and $\sigma_\beta = 0.59$ cases/day. The small value of $\mu_\beta^2/\sigma_\beta^2 = 0.096$, equivalent to the dispersion parameter [16, 32, 33], provides clear evidence for superspreading during early stages of the COVID-19 pandemic in the United States. (See S7 Appendix for discussion about defining the dispersion parameter in terms of the daily infection rate).

These results for $\mu_\beta$ and $\sigma_\beta$ can be used to further quantify the extent of superspreading under the assumption that $p(\beta)$ follows a gamma distribution (as in Ref. [18]). In the Methods section we present a derivation of the cumulative share of infections, $Y$, caused by the top $X$ portion of most infectious cases. The corresponding "Lorenz curve" $Y(X)$ is plotted in Fig 3. This result implies (using our relatively conservative estimate of $\sigma_\beta$) that 81% of new infections

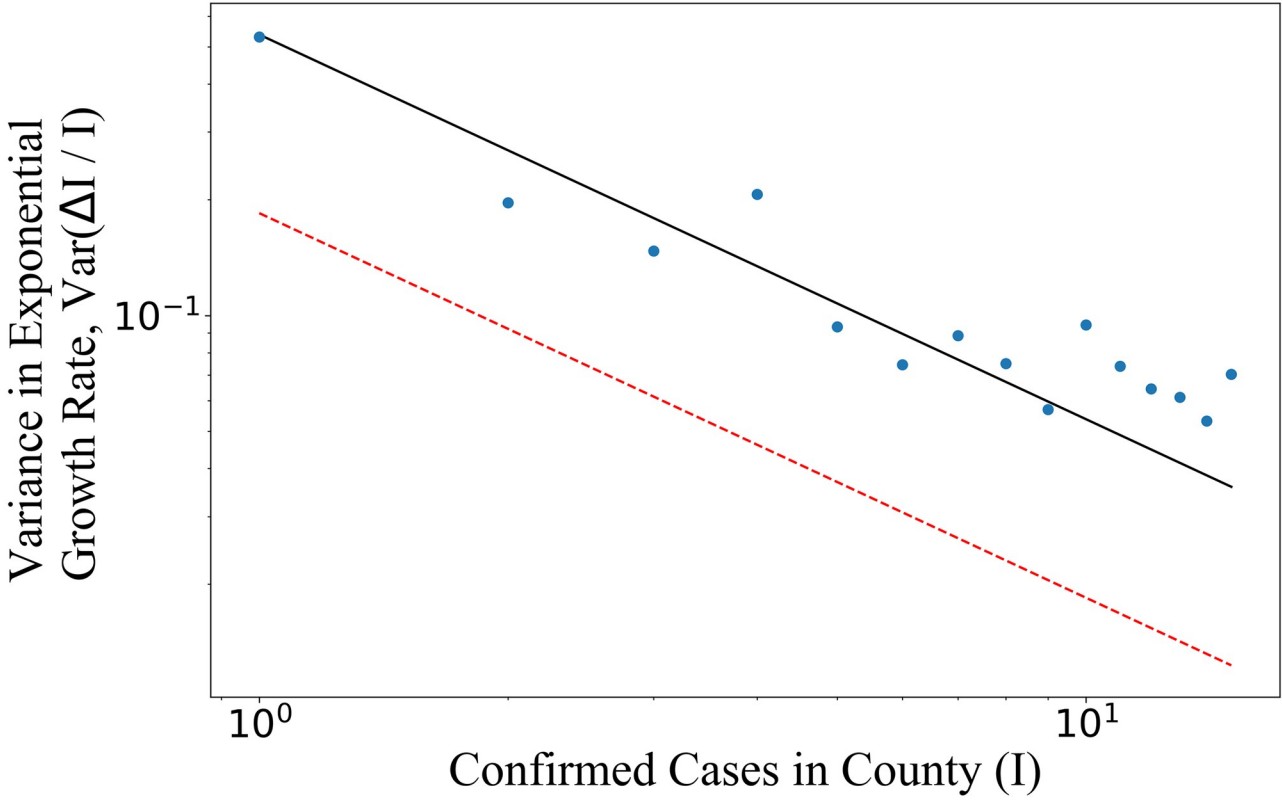

**Fig 2. As the number of infections $I$ in a given county increases, the variance in exponential growth rate, Var($\Delta I/I$), decreases as $(\mu_\beta + \sigma_\beta^2)/I$.** Each data point at a given $I$ is calculated by taking the sample variance in $\Delta I/I$ across all counties when they have $I$ cases. We observe that the USA data (blue) is inconsistent with a model of uniform infectiousness, or $\sigma_\beta = 0$ (dashed red line). A fit to the data (solid black line) implies a large variance in infectiousness, such that $\sigma_\beta/\mu_\beta \gtrsim 3.2$.

are produced by the top 10% of most infectious individuals, while only about 4.5% of cases arise from the 80% of infected individuals with the lowest infection rates.

## Discussion

As we have shown, a wide distribution $p(\beta)$ in infectiousness $\beta$ leads to large statistical variation in the early-time growth rate of a disease. By calculating the variance in growth rate among different subpopulations one can infer the variance in $p(\beta)$. Our result for COVID-19 cases in the USA suggests that $\sigma_\beta/\mu_\beta \gtrsim 3.2$, implying a relatively severe superspreading. If we further assume that $p(\beta)$ follows a gamma distribution (as in Ref. [18]), then we can produce a more direct estimate of the extent of superspreading (Fig 3). Our relatively simple and direct method, based on a calculation of variance in reported case data, can be contrasted with more complicated methods for inferring the dispersion parameter that are based on maximum likelihood estimation (e.g., Ref. [33] develops such a method using simulated data), cluster size distributions [16, 34], and surveillance or tracing data [14, 15]. These methods also tend to yield a lower-bound estimate for $\sigma_\beta/\mu_\beta$. While studies based on testing and contact tracing (e.g., Refs. [18, 35–37]) remain the definitive method for assessing superspreading, the method we present here may provide a much simpler way of estimating its prevalence across a much larger population.

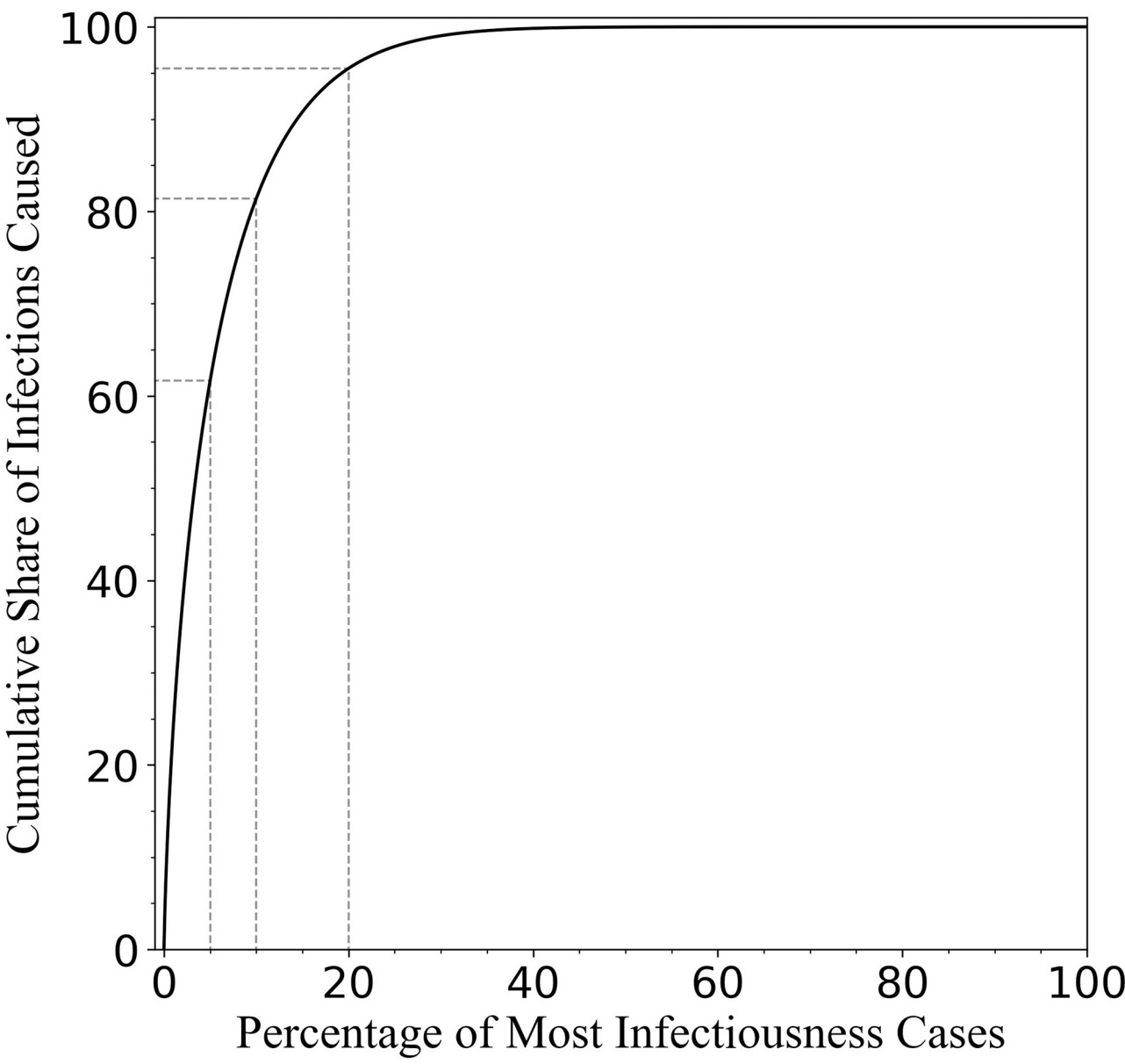

**Fig 3. An estimated Lorenz curve for SARS-CoV-2 infections in the USA, which displays the percentage of new cases that are caused by a given cumulative percentage of most infectious individuals (solid black).** A few points in the curve are highlighted (dashed grey lines): 61.7%, 81.4%, and 95.5% of new cases are caused by the top 5%, 10%, and 20% infectious cases, respectively. Accounting for undetected and asymptomatic cases would apparently make this curve steeper, corresponding to more severe superspreading.

We emphasize that our analysis is unable to determine whether this large variance is a result of differing biological symptoms, social behavior, or other possible explanations. Additionally, this estimation is carried out for early times to minimize effects from a time varying $p(\beta)$ and therefore predominantly speaks to the infectiousness prior to widespread lockdown measures.

We close by commenting on a number of complicating factors that we did not include in our analysis and which, one might suspect, could alter our primary finding of a large value of $\sigma_\beta/\mu_\beta$. For example, we have assumed a uniform value of $\mu_\beta$ across different geographic locations; we have neglected undetected cases; we have ignored the possible variation in detection

rate $p_{\text{det}}$ among different counties; we have effectively treated each county as an isolated population and have neglected cross-county interactions; and we have ignored the effects of the incubation period as well as the potential variation in incubation periods between individuals. In the Supplemental Information, we consider each of these mechanisms in turn and show that none of them can explain our result, so that our conclusion of prevalent superspreading of SARS-CoV-2 in the USA remains robust. In brief: the variation in $\mu_\beta$ among different geographic locations is too small to explain the observed variance in growth rate [S2 Appendix]; neglecting undetected cases leads to an *underestimate* of the variance $\sigma_\beta^2$, so that our result is effectively a lower bound for the prevalence of superspreading [S3 Appendix]; variation in $p_{\text{det}}$ between counties does not directly affect the variance in the growth rate $(\Delta I_{\text{det}})/I_{\text{det}}$, other than to provide an average of $p_{\text{det}} < 1$, which results in a lower-bound estimate of $\sigma_\beta^2$ [S4 Appendix]; cross county interactions tend to reduce the variance, so our result cannot be explained as a consequence of such interactions [S5 Appendix]; and variations in incubation period can only reduce the apparent variance in growth rate [S6 Appendix].

## Methods

### Data source

We use publicly available data taken from the data set provided by the Center for Systems Science and Engineering (CSSE) at Johns Hopkins University [29] to estimate $\mu_\beta$. Knowing $\mu_\beta$ enables us to determine $\sigma_\beta$ by taking a best fit to Eq (6). Counties that recorded $\Delta I < 0$ at any point are discarded from the analysis due to the potential for recording error; such counties comprise $\sim$20% of all counties.

### Numerical simulation

We corroborate Eqs (5) and (6) using a numerical simulation of the trajectories of infection growth, $I(t)$, for a given distribution $p(\beta)$. Reference 18 has suggested that infectiousness follows a gamma distribution, and consequently, $P(n) = \text{NB}(n; \mu_\beta^2/\sigma_\beta^2, \mu_\beta/(\mu_\beta + \sigma_\beta^2))$ where NB is the negative binomial distribution [10, 16]. Using this assumption, we simulate the growth of the epidemic by assuming that a given individual $i$, with infectiousness $\beta_i$ that is drawn randomly from $p(\beta)$, generates a number $n_i$ of new cases each subsequent day that is drawn from $\text{Pois}(n_i; \beta_i)$. The simulation results confirm Eqs (5) and (6), as shown in S1 Appendix. Numerical simulations were performed using Python; the primary analysis is publicly available [38] and the simulations are available upon request to the corresponding author.

### Derivation of the curve $Y(X)$

Following Ref. [18], we assume that the distribution of infectiousness, $p(\beta)$, follows a gamma distribution. This assumption also allows us to further quantify the degree of superspreading by deriving a mathematical relation for the curve $Y(X)$, where $Y$ represents the proportion of infections produced by the top $X$ fraction of most infectious individuals. In particular, one can calculate the fraction of individuals $X_{\beta_0}$ with infectiousness larger than a given value $\beta_0$, as well as the fraction of secondary infections $Y_{\beta_0}$ that these individuals are expected to cause:

$$X_{\beta_0} = \int_{\beta_0}^{\infty} d\beta\, p(\beta) = Q\left(\frac{\mu_\beta^2}{\sigma_\beta^2}, \beta_0 \frac{\mu_\beta}{\sigma_\beta^2}\right) \tag{7}$$

$$Y_{\beta_0} = \int_{\beta_0}^{\infty} d\beta\, p(\beta) \frac{\beta}{\mu_\beta} = Q\left(1 + \frac{\mu_\beta^2}{\sigma_\beta^2}, \beta_0 \frac{\mu_\beta}{\sigma_\beta^2}\right), \tag{8}$$

where $Q$ is the Regularized Gamma function. By eliminating $\beta_0$ we find

$$Y = Q\left(1 + \frac{\mu_\beta^2}{\sigma_\beta^2}, Q^{-1}\left(\frac{\mu_\beta^2}{\sigma_\beta^2}, X\right)\right). \tag{9}$$

Fig 3 displays the cumulative share of infections, $Y$, caused by the top $X$ portion of most infectious cases.

## Supporting information

**S1 Appendix. Simulations.**
(PDF)

**S2 Appendix. Variance in $\mu_\beta$.**
(PDF)

**S3 Appendix. Undetected cases.**
(PDF)

**S4 Appendix. Variance in testing.**
(PDF)

**S5 Appendix. Cross-county interactions.**
(PDF)

**S6 Appendix. Variance in incubation period.**
(PDF)

**S7 Appendix. Dispersion parameter comparison.**
(PDF)

## Acknowledgments

The authors are grateful to N. E. Skinner for helpful conversations.

## Author Contributions

**Conceptualization:** Brian Skinner.

**Data curation:** Calvin Pozderac.

**Formal analysis:** Calvin Pozderac.

**Investigation:** Calvin Pozderac, Brian Skinner.

**Methodology:** Calvin Pozderac, Brian Skinner.

**Project administration:** Brian Skinner.

**Software:** Calvin Pozderac.

**Supervision:** Brian Skinner.

**Visualization:** Calvin Pozderac.

**Writing – original draft:** Calvin Pozderac.

**Writing – review & editing:** Calvin Pozderac, Brian Skinner.

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
