## [Decision Letter · Decision Letter 0]

27 Jan 2021

PONE-D-20-30570

Superspreading of SARS-CoV-2 in the USA

PLOS ONE

Dear Dr. POZDERAC,

Thank you for submitting your manuscript to PLOS ONE. After careful consideration, we feel that it has merit but does not fully meet PLOS ONE’s publication criteria as it currently stands. Therefore, we invite you to submit a revised version of the manuscript that addresses the points raised during the review process.

Your manuscript was reviewed by 2 experts in the field. Both reviewers identified many important issues in your submission which require your careful attention. Please review the attached comments and provide point-by-point responses.

We look forward to receiving your revised manuscript.

Kind regards,

Yury E Khudyakov, PhD

Academic Editor

PLOS ONE

Journal Requirements:

Reviewers' comments:

Reviewer's Responses to Questions

**Comments to the Author**

1. Is the manuscript technically sound, and do the data support the conclusions?

Reviewer #1: Partly

Reviewer #2: Yes

2. Has the statistical analysis been performed appropriately and rigorously? 

Reviewer #1: No

Reviewer #2: Yes

3. Have the authors made all data underlying the findings in their manuscript fully available?

Reviewer #1: Yes

Reviewer #2: Yes

4. Is the manuscript presented in an intelligible fashion and written in standard English?

Reviewer #1: Yes

Reviewer #2: Yes

5. Review Comments to the Author

Reviewer #1: In this study, Calvin and colleagues introduced a novel method to determine the variation of infectiousness among populations based on SIR model. And then authors used this method to illustrate there was significant variations of infectiousness among populations in USA known as superspreading events. The study is interesting and the result (superspreading events in USA) will also contribute to tailor the prevention and control policies for COVID-19 epidemic in USA. However, I have several concerns about this study.

The COVID-19 has highly varied incubation period (with mean of 5.2 days, and the distribution of incubation period was estimated as 12.5 days). The highly variable incubation time would result in surge for the number of confirmed cases in some specific times. As the method only used the number of confirmed case each day, it is important to know how the highly variable incubation time affect the result? As a novel method, it should be first tested in pervious data (such as SARS-CoV, Ebola, ZIKA etc) to illustrate the accuracy of the method. Then the result for COVID-19 would be more convincing.

Reviewer #2: In this article, the authors derive a relationship between the variance in epidemic growth rates and current size of the epidemic, formalizing the intuition that super spreading has the greatest impact on epidemic dynamics when the number of infected individuals is small. They then apply this relationship to COVID-19 case data from the U.S. to estimate superspreading of SARS-CoV-2.

Major comments

1. The authors compare their estimate of $\\mu_k^2/\\sigma_k^2$ base on mean and standard deviation of daily transmission rate to the dispersion parameters presented in the literature which are instead based on duration of infection transmission rate. These numerical estimates therefore seem incomparable without further transformation. For example, if the daily transmission rate was Gamma distributed with mean $\\mu_k$ and variance $\\sigma_k^2$ and each individual’s daily transmission rate was a independent draw from this distribution, I believe their duration of infection transmission rate (based on the manuscript’s assumption of a 14 day duration of infection) would have a distribution with the ratio of mean^2/variance being $14 \\mu_k^2/\\sigma_k^2$.

2. The authors point out that their estimates provide a lower bound on $sigma_k$ use this to identify prevalent superspreading throughout the USA. Given that superspreading of SARS-CoV-2 is generally accepted and the effort taken by the authors to quantify the effects of different assumptions, it might be more useful to use this methods of measuring superspreading to identify a range of possible values of $\\sigma_k$ based on plausible assumptions, that is, identify both lower and upper bounds on $\\sigma_k$, and compare this to the existing estimates in the literature.

3. The authors address many potential complications that could impact their findings, including varying testing rates in different counties (Appendix S4). A further complication would be differentially increasing testing rates, where some counties ramped up testing faster than others. Similarly, the estimates might be impacted by day of week patterns in reported cases, where weekly cases often show cyclical patterns indicating that testing and reporting of confirmed cases is variable based on the day of the week. While I doubt either of these would substantially affect the major findings of the paper, some discussion of the potential impact of this would be useful.

Minor comments

1. The choice of k to denote the transmission rate seem to be a likely source of confusion as k is frequently used as the dispersion parameter in literature on superspreading (e.g., see refs 10, 16, 18 from the manuscript).

2. Figure 1: Further details should be provided as to the set of counties and time periods included in part B of the figure.

3. Line 94: The authors state that the recovery period for COVID-19 is 14 days, but other literature suggest that viral shedding may decline and seroconversion may occur 7 days after symptom onset (e.g. Wölfel et al. 2020 Virological assessment of hospitalized patients with COVID-2019. Nature. https://doi.org/10.1038/s41586-020-2196-x). While this is negligible in the scheme of affecting the portion of the population susceptible, it nevertheless would be more accurate to provide a range here.

4. Figure 2: The caption for could be made clearer by indicating that the blue points are bucketed data where buckets are determined by confirmed cases. Additionally, clarification should be made as to how the number of confirmed cases was determined (at the end of the 14 day period?). Finally, the y-axis in Figure 2 needs additional labels to be informative.

5. I would strongly encourage the authors to make the code publicly available.

6. PLOS authors have the option to publish the peer review history of their article (what does this mean?). If published, this will include your full peer review and any attached files.

Reviewer #1: No

Reviewer #2: No

---

## [Author Response · Author response to Decision Letter 0]

4 Mar 2021

Our responses to the reviewers and editor's comments can be found in our attached 'Response to Reviewers.'

---

## [Editor Report · Decision Letter 1]

8 Mar 2021

Superspreading of SARS-CoV-2 in the USA

PONE-D-20-30570R1

Dear Dr. POZDERAC,

We’re pleased to inform you that your manuscript has been judged scientifically suitable for publication and will be formally accepted for publication once it meets all outstanding technical requirements.

Kind regards,

Yury E Khudyakov, PhD

Academic Editor

PLOS ONE
---

## [Editor Report · Acceptance letter]

10 Mar 2021

PONE-D-20-30570R1 

Superspreading of SARS-CoV-2 in the USA 

Dear Dr. Pozderac:

I'm pleased to inform you that your manuscript has been deemed suitable for publication in PLOS ONE. Congratulations! Your manuscript is now with our production department. 

Kind regards, 

on behalf of

Dr. Yury E Khudyakov 

Academic Editor

PLOS ONE